# Properties of Compacts from Mixtures of Calcium Fluoride Micro- and Nanopowders

**Vladislav G. Ilves** [1,*], **Sergey Yu. Sokovnin** [1,2], **Sergey V. Zayats** [1] and **Mikhail G. Zuev** [3]

1  Institute of Electrophysics, Ural Branch, Russian Academy of Sciences, 620016 Yekaterinburg, Russia
2  Institute of Physics and Technology, Ural Federal University Named after the First President of Russia B.N. Yeltsin, 21 Mira Street, 620002 Yekaterinburg, Russia
3  Institute of Solid State Chemistry, Ural Branch, Russian Academy of Sciences, 620990 Yekaterinburg, Russia
*  Correspondence: ilves@iep.uran.ru

**Abstract:** In this work, compacts from mechanical mixtures of $CaF_2$ micron powder (MCP) with $CaF_2$ nanopowder (NP) additives were produced, with mass ratios of the mixture components ranging from 10:0.125 to 10:1, respectively, using magnetic pulse (MP) and static pressing (SP) methods. The effects of pressure ($P_p$) and pressing temperature ($T_p$), concentration and phase composition of the additive on the density and color of compacts were studied, taking into account the properties of the initial components of the mixtures. The evolution of pulsed cathodoluminescence (PCL) spectra and photoluminescence (PL) of compacts from pure powders and their mixtures depending on $P_p$, $T_p$ and characteristics of initial $CaF_2$ NP was also studied. A new near-infrared (NIR) band associated with fluoride vacancies was discovered with a maximum at ~765 nm in PCL spectra of compacts produced by MP at a temperature of 425 °C. A blue band at 435 nm associated with impurity oxygen vacancies in the $CaF_2$ lattice was found in PL spectra compacts of pure NP and powder mixtures. The density of compacts of pure NP and MCP reached 89% of the theoretical density, and the density of compacts of mixtures did not exceed 78%. The defective structure and phase composition of the $CaF_2$ NP had a decisive effect on the luminescent properties of compacts from mixtures of micro- and nanopowders.

**Keywords:** compacts from mixtures of micro- and nanopowders of $CaF_2$; magnetic pulse and static pressing; pulsed cathodoluminescence; photoluminescence

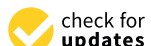



## 1. Introduction

Achievements in the development of oxide optical ceramics based on REM oxides and yttrium–aluminum garnet [1–4] revived the interest in optical fluoride nanoceramics at the beginning of the 21st century [5–10]. In the synthesis of fluoride nanoceramics, two approaches are used—"bottom-up" and "top-down", each of which has several options for realization. The two methods most widely used for the synthesis of fluoride optical ceramics are a hot pressing method and a deep plastic deformation method (hot forming) [5,6,11,12]. In the hot pressing method, powder charge is simultaneously exposed to temperature (0.5–0.8 $T_{melting}$) and pressure (up to 300 MPa), which makes it possible to reduce sintering temperature [13,14]. In hot pressing, the sample heated to a certain temperature undergoes plastic deformation under pressure [15]. The microstructures of ceramics produced by both methods are similar. Recently, the method of spark plasma sintering (SPS) has become widespread; however, when using it, there are great problems with contamination of the final product with graphite [16–19].

Due to the peculiarities of the chemical behavior of fluorides, the technologies for producing fluoride nanoceramics differ significantly from the technologies for producing oxide ceramics. In particular, it is necessary to use vacuum chambers to prevent pyrohydrolysis [5,20]. Particular attention is paid to the selection of precursors, for which both commercial fluoride powders and powders produced under laboratory conditions are used. The main requirements for precursors are high chemical purity, minimal porosity

and choice of the method of synthesis of the precursor itself [5,21]. It is also necessary to use graphite in hot pressing or growing single crystals since fluoride melts do not wet and do not interact with the graphite crucible.

Recently, the authors of this paper used a relatively new method of pulsed electron evaporation in vacuum (PEBE) [22,23] to produce nanopowders (NPs) of alkaline earth difluorides $CaF_2$ [24] and $BaF_2$ [25] and trifluoride $CeF_3$ [26]. Simple evaporation of the target is a favorable condition for the synthesis of fluoride nanopowders in the PEBE method (tablets pressed from any high-purity commercial fluoride powder) in a graphite crucible under vacuum conditions (or inert gases, e.g., argon) followed by vapor deposition on a cold substrate (such a substrate may be any thermally resistant and chemically inert to a particular fluoride material, such as graphite or conventional window glass). The chemical purity of the final product (nanopowder) depends solely on the chemical purity of the starting material and can be even higher than that of the commercial powder.

During evaporation in a vacuum, the presence of a reduced metal in the final product (a matrix cation, which can have both a positive (increase in compact density) and a negative effect (formation of an oxide shell on the surface of NPle metals) on the properties of the compact) is possible. The compaction of fluoride NPs containing metallic NPles is very poorly understood [27,28]. $CaF_2$ powders kept in vapors of Ca (so-called additive coloring [29–31]) have some degree of similarity to such powders; however, the mechanism of formation of such powders considerably differs from the mechanism of formation of $CaF_2$ NP produced by the PEBE method in a vacuum.

The advantage of the NPle fluorides produced by the PEBE method is the relatively small NPle size, from 3 to 15 nm. This allows, if necessary, removing undesirable moisture and volatile organic compounds adsorbed by the developed surface of the nanopowder and varying the size of the nanocrystals by simple thermal annealing in a vacuum or inert atmosphere.

However, the high specific surface area and mesoporous type of fluoride powders are a strong obstacle to using them for the manufacture of optically transparent nanoceramics. Nevertheless, the porosity of such nanopowders is interparticular because the pores contain aggregates and agglomerates of nanoparticles rather than the nanoparticles themselves having a quasi-spherical shape. The interparticular pores may partially disappear during compact pressing and recrystallization when the compact is further sintered in a vacuum or argon in an electric furnace. At the annealing temperature of 400–450 °C, there is no noticeable growth of fluoride nanocrystal grains [24–26].

The purpose of the present work was to investigate the effect of small $CaF_2$ NP additives on the density and luminescent properties of compacts made from mechanical mixtures of pure nano- and micropowders of $CaF_2$ using static (SP) and magnetic pulse pressing (MP) methods to assess the potential use of such compacts in various applied fields of science and technology.

## 2. Experimental Section

### 2.1. CaF₂ Powders

#### 2.1.1. Commercial CaF₂

Commercial $CaF_2$ micron powder of OSCH grade (TU 6-09-2412-84) was used directly for sintering without any pre-treatment of pure calcium fluoride nanopowder (NP). Mass fraction CaF2 was not less than 98%. Mass fraction of impurities was not more than: Ba, $1 \times 10^{-3}$; Fe, $5 \times 10^{-4}$; Si, $5 \times 10^{-3}$; Cu, $5 \times 10^{-5}$; Pb, $5 \times 10^{-4}$.

#### 2.1.2. Synthesis of CaF₂ NP Using PEBE

$CaF_2$ NP was produced by pulsed electron evaporation (PEBE) under vacuum (residual pressure 4 Pa) [22,23]. A detailed description of the production process and basic physicochemical characteristics of $CaF_2$ NP produced by the evaporation of compacts made of micron $CaF_2$ are described in detail in our early work [24].

## 2.2. Fabrication of CaF$_2$ Ceramics

CaF$_2$ NP produced by PEBE was used as an additive to the matrix (commercial micron powder) in the preparation of the compact starting mixtures using magnetic pulse pressing (MP) methods at 425 °C and room temperature (RT) and static pressing (SP) at RT. Mixing of micro- and nanopowders was performed in a dry manner using a mortar and pestle at the following powder weight ratios: 10:0.125, 10:0.25, 10:0.5 and 10:1. The compacts were pressed in a collapsible mold consisting of three shells. The matrix channel was 5 mm. Pressing was carried out on two uniaxial presses: a static press PRG-70 and a magnetic pulse. During SP, the pressure varied between 800 and 2000 kgf. During MP, the pressure did not exceed 1 GPa at a capacitor bank voltage of 0.8 kV. Static pressing was carried out in air; additional vacuum degassing at room temperature for 60 min was used for MP. The geometric dimensions of the samples were measured with a standard micrometer (error ± 2 μm). The weight of the samples was measured on a standard balance with an error of ±0.001 g.

## 2.3. Characterizations

Diffractograms were taken on a D8 DISCOVER diffractometer on copper radiation (Cu K$\alpha$1,2 $\gamma$ = 1542 Å) with a graphite monochromator on a diffracted beam. Processing was performed using TOPAS 3. When estimating the average size of crystallites, a correction factor K (in the Scherer formula) = 0.8 was used. To analyze the textural characteristics of fluorides degassed at 373 K for one hour, nitrogen adsorption–desorption isotherms were recorded using a Micromeritics TriStar 3000 V6.03 A adsorption analyzer. The specific surface area (SSA) was calculated according to the standard Brunauer, Emmett and Teller method. The total pore volume Vp was evaluated from the nitrogen adsorption at $p$/p0 $\approx$ 0.99, where $p$ and p0 denote the equilibrium and saturation pressure of nitrogen at 77.4 K, respectively. The measurement error of the SSA is ±(5...10)%.

PL spectra were registered by means of an MDR-204 spectrometer while exciting the samples using a DDS-30 deuterium lamp and an R928 photomultiplier from Hamamatsu. When recording PL spectra, a glass filter with a bandwidth of 0.6–1 μm was used to suppress second orders in the spectra. Smoothing of noisy spectral signals was carried out. Pulsed cathodoluminescence (PCL) was excited and studied using the KLAVI setup [32–34]. Before the study, the samples were not subjected to special treatment and were irradiated in air with an electron beam (pulse duration of 2 ns, current density of 130 A/cm$^2$, average electron energy of 170 keV). Time-integrated emission spectra were recorded by two independent multichannel photodetectors (based on CCD arrays) in the ranges 250–380 and 400–870 nm at an exposure of 50 ms. The spectral information was averaged over 40 pulses, while the stability of the amplitude parameters of the PCL spectrum was no less than 90%. The wavelength measurement error did not exceed ±0.5 nm.

## 3. Results and Discussion

Some characteristics of micron and NP CaF$_2$ mixes' components are given in Figure 1.

XRD analysis (Figure 1a) showed that the initial CaF$_2$ NP contained two crystalline phases: cubic fluoride phase, S.G.: Fm-3 m (225), PDF No. 00-035-0816 card, 54,631 $\rho$ = 3.181(4) g/cm$^3$, content 95%, R$_b$ (%) = 3.024, and tetragonal fluoride phase, S.G.: P4/mmm (123), PDF No. 01-070-2739, $\rho$ = 3.23(2) g/cm$^3$, content < 5%, R$_b$ (%) = 8.594, R$_{exp}$ = 15.32, R$_{wp}$ = 18.33.

The micron powder also consisted of cubic and tetragonal fluoride phases: S.G.: Fm-3 m (225), PDF No. 00-035-0816, a = 54,631 Å $\rho$ = 3181 g/cm$^3$, content > 96 wt.%, R$_b$ = 3.061, and S.G.: P4/mmm (123), PDF No. 01-070-2739, a = 3735 Å, c = 2867 Å, $\rho$ = 3242 g/cm$^3$, content $\approx$ 3 wt.%, R$_b$ = 15.55, respectively.

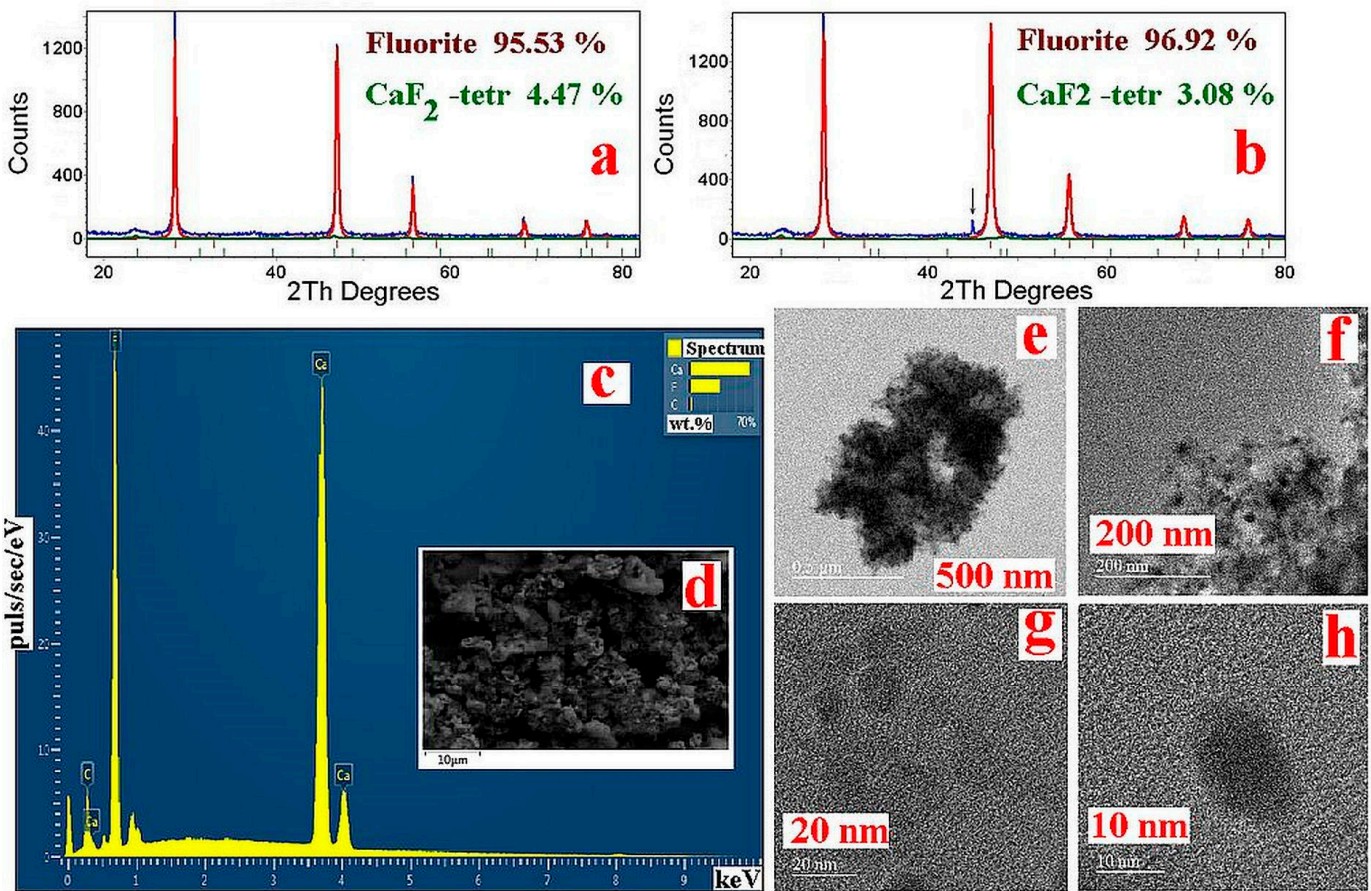

**Figure 1.** XRD patterns of the commercial (**a**) and synthesized (**b**) $CaF_2$ powders; EDX spectra (**c**) and SEM images (**d**) of $CaF_2$ NP; TEM (**e–g**) and HRTEM (**h**) images of the synthesized $CaF_2$ powders. TEM-HRTEM images recorded by M. Rähne, an employee of the Institute of Physics at the University of Tartu (Estonia) [24].

An unidentified phase was also present, with coherent scattering regions (CSRs) >> 200 nm, accounting for <1% by volume; the phase is not a [Ca + F] compound. It is impossible to determine the crystalline phase from a single reflection (the reflection is indicated by an arrow in Figure 1b). The dimensions of the CSR and the lattice periods of the cubic phase of the commercial powder and NP are shown in Table 1.

**Table 1.** Lattice parameters and CSR in cubic and tetragonal phases of commercial calcium fluoride powder and calcium fluoride nanopowder.

| | $CaF_2$ commercial | | | | | |
|---|---|---|---|---|---|---|
| CSR, nm | $CaF_2$ cub Period, Å | $\rho$, g/cm³ | CSR, nm | $CaF_2$ tetr Period, Å | $\rho$, g/cm³ |
| | $5.465 \pm 0.003$ | $3.177 \pm 0.004$ | $14 \pm 2$ | a = $3.784 \pm 0.007$ c = $2.604 \pm 0.006$ | $3.48 \pm 2$ |
| | $CaF_2$ initial | | | | | |
| CSR, nm | $CaF_2$ cub Period, Å | $\rho$, g/cm³ | CSR, nm | $CaF_2$ tetr Period, Å | $\rho$, g/cm³ |
| $45 \pm 3$ | $5.463 \pm 0.003$ | $3.181 \pm 0.004$ | $\approx 14$ | a = $3.734 \pm 0.007$ c = $2.875 \pm 0.009$ | $3.23 \pm 2$ |

The morphology and elemental composition of the $CaF_2$ NP are shown in Figure 1c,d. The transmission electron microscopy images of the $CaF_2$ NP are shown in Figure 1e,f [24]. Figure 1e shows an NPle CaF2 agglomerate having a size of about 500–600 nm, typical of

most NPles (fluorides, oxides, sulfides and other compounds) produced by PEBE. Such an agglomerate is a 3D spatial structure consisting of aggregates several tens of meters in size (Figure 1f), which in turn (Figure 1g,h) consist of individual NPles of a quasi-spherical form, about 10–15 nm in size, with a relatively weak agglomeration degree.

### 3.1. Textural Analysis

The textural properties of the commercial micron powder from which the target and the initial sample of NP (sample S0) produced by the PEBE method were made are shown in Figure 2.

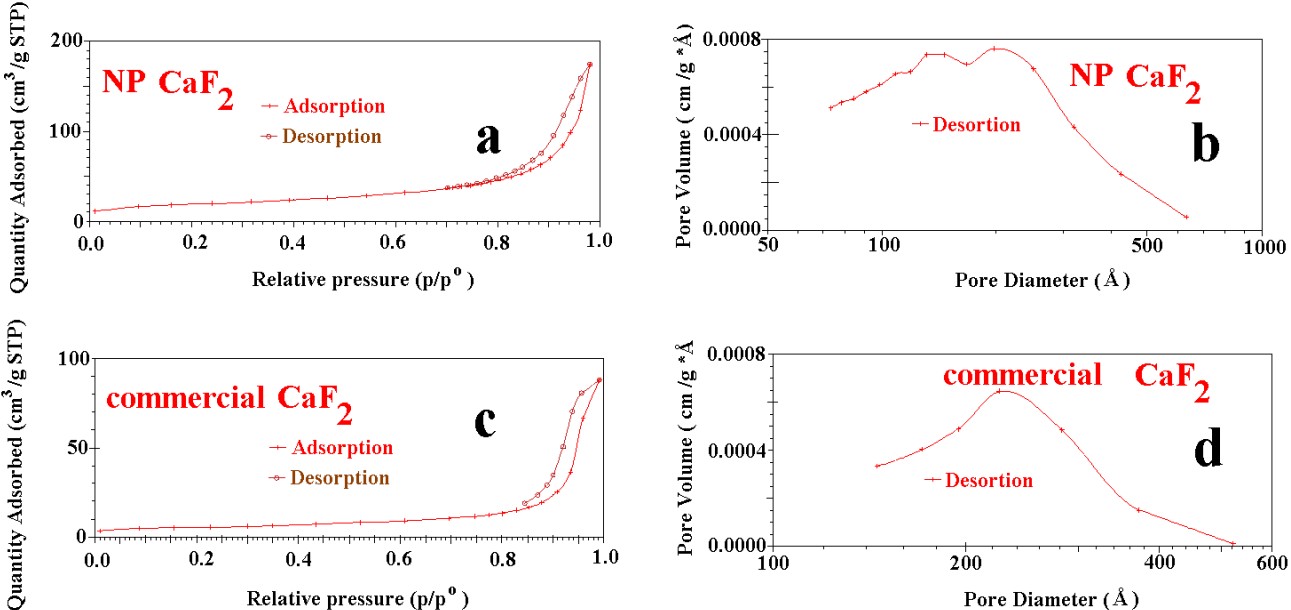

**Figure 2.** Nitrogen adsorption–desorption isotherms (**a**,**c**) and pore size distribution curves (**b**,**d**) of initial CaF$_2$ NP and commercial CaF$_2$ powder.

Both powders were mesoporous powders, as indicated by the presence of a type IV hysteresis loop on their nitrogen adsorption–desorption isotherms. The micron powder showed unimodal pore size distribution, while the NP S0 showed multimodal pore distribution due to its spatial structure of nanoparticle (NPle) aggregates (see Figure 1d,e). There was also a very strong deviation from the stoichiometry towards calcium Ca: F = 0.74–1.1 (at.%) (EDX spectra were recorded in three sections of sample S0), which confirmed the possible presence of metal Ca nanoparticles in the NP. The spatial structure of the aggregates and the quasi-spherical form of the individual NPles in CaF$_2$ NP confirm the TEM and HRTEM images presented in Figure 1e,f, respectively.

Table 2 shows the textural properties of the calcium fluoride nanopowder and commercial calcium fluoride powder used in the preparation of the initial compaction mixtures.

**Table 2.** Textural properties of components of commercial powder and CaF$_2$ nanopowder mixtures.

| Composition | Powder Type | SSA, m$^2$/g | Total Pore Volume, cm$^3$/g | Average Pore Diameter, (nm) |
|---|---|---|---|---|
| CaF$_2$ | Commercial | 18.5 | 0.13 | 25 |
| CaF$_2$ | Nano | 64.3 | 0.25 | 21 |

### 3.2. Compact Density Analysis

Table 3 shows the density of compacts from mixes of CaF$_2$ micro- and nanopowders, as well as the pressing parameters of the MP and SP methods.

**Table 3.** Density of $CaF_2$ compacts and parameters of their pressing by SP and MP methods.

| Sample | Share of Nanopowder | Degassing | | Pressing | | Release | | Compact | | | |
|---|---|---|---|---|---|---|---|---|---|---|---|
| | | $T_d$, °C | $t_d$, min | $S_p$, kgf ($U_c$, kV) | $T_p$, °C | $T_r$, °C | $t_r$, min | m, g | Ø, mm | d, mm | ρ, g/cm³ |
| 5941SP | 0 | RT | - | 800 kgf | RT | - | - | 0.052 | 4.62 | 1.72 | 1.80 |
| 5942SP | 0 | RT | - | 1600 kgf | RT | - | - | 0.059 | 4.62 | 1.51 | 2.33 |
| 5943SP | 0 | RT | - | 2000 kgf | RT | - | - | 0.048 | 4.62 | 1.37 | 2.09 |
| 5944SP | 1.25 | RT | - | 800 kgf | RT | - | - | 0.057 | 4.60 | 1.65 | 2.08 |
| 5945SP | 1.25 | RT | - | 1600 kgf | RT | - | - | 0.035 | 4.62 | 1.04 | 2.00 |
| 5946SP | 1.25 | RT | - | 2000 kgf | RT | - | - | 0.080 | 4.62 | 1.98 | 2.41 |
| 5947SP | 2.5 | RT | - | 800 kgf | RT | - | - | 0.052 | 4.60 | 1.52 | 2.05 |
| 5948SP | 2.5 | RT | - | 1600 kgf | RT | - | - | 0.039 | 4.61 | 1.14 | 2.05 |
| 5949SP | 2.5 | RT | - | 2000 kgf | RT | - | - | 0.062 | 4.62 | 1.64 | 2.26 |
| 5950SP | 5 | RT | - | 800 kgf | RT | - | - | 0.066 | 4.61 | 1.86 | 2.12 |
| 5951SP | 5 | RT | - | 1600 kgf | RT | - | - | 0.065 | 4.63 | 1.72 | 2.25 |
| 5952SP | 5 | RT | - | 2000 kgf | RT | - | - | 0.073 | 4.63 | 1.95 | 2.22 |
| 5953MP | 0 | 425 | 120 | (0.8) | 425 | 425 | 60 | 0.049 | 4.53 | 1.08 | 2.82 |
| 5954SP | 10 | RT | - | 800 kgf | RT | - | - | 0.041 | 4.61 | 1.24 | 1.99 |
| 5955SP | 10 | RT | - | 1600 kgf | RT | - | - | 0.046 | 4.62 | 1.48 | 1.86 |
| 5956SP | 10 | RT | - | 2000 kgf | RT | - | - | 0.055 | 4.62 | 1.48 | 2.21 |
| 5957SP | 100 | RT | - | 800 kgf | RT | - | - | 0.054 | 4.59 | 1.60 | 2.04 |
| 5958SP | 100 | RT | - | 1600 kgf | RT | - | - | 0.076 | 4.60 | 1.71 | 2.68 |
| 5959SP | 100 | RT | - | 2000 kgf | RT | - | - | 0.059 | 4.61 | 1.32 | 2.68 |
| 5960MP | 1.25 | 425 | 120 | (0.8) | 425 | 425 | 60 | 0.055 | 4.50 | 1.63 | 2.12 |
| 5961SP | 2.5 | 425 | 120 | (0.8) | 425 | 425 | 60 | 0.061 | 4.53 | 1.42 | 2.66 |
| 5962MP | 0 | RT | 60 | (0.8) | RT | - | - | 0.058 | 4.62 | 1.53 | 2.26 |
| 5963MP | 1.25 | RT | 60 | (0.8) | RT | - | - | 0.054 | 4.62 | 1.31 | 2.45 |
| 5964MP | 2.5 | RT | 60 | (0.8) | RT | - | - | 0.052 | 4.62 | 1.35 | 2.30 |
| 5965MP | 5.0 | RT | 60 | (0.8) | RT | - | - | 0.052 | 4.61 | 1.37 | 2.28 |
| 5966MP | 5.0 | 425 | 120 | (0.8) | 425 | 425 | 60 | 0.072 | 4.50 | 1.77 | 2.56 |
| 5967MP | 10 | RT | 60 | (0.8) | RT | - | - | 0.045 | 4.62 | 1.16 | 2.31 |
| 5968MP | 100 | RT | 60 | (0.8) | RT | - | - | 0.058 | 4.60 | 1.36 | 2.57 |
| 5969MP | 10 | 425 | 120 | (0.8) | 425 | 425 | 60 | 0.059 | 4.50 | 1.33 | 2.79 |
| 5970MP | 100 | 425 | 120 | (0.8) | 425 | 425 | 60 | 0.048 | 4.53 | 1.38 | 2.15 |

Designations: RT—room temperature; sp—static pressure; $t_d$—degassing time; $t_r$—release time; $U_c$—charging voltage; $T_d$, $T_p$ and $T_r$—degassing, pressing and release temperature, respectively. Compact options: m—mass, Ø—diameter, d—thickness, ρ—density. Subscripts SP and MP for sample numbers are given only in Table 3 for clarity.

Table 2 shows that with a comparable pore diameter for micro- and nanopowders, the total pore volume in the NP is almost twice that of the micron powder. A large pore volume usually negatively affects the density of compacts made from NP [35]; however, when using SP, a greater density of NP compacts was achieved (5957–5959, Table 3) in the entire pressure range in comparison with the density of micron powder compacts (5941–5943, Table 3). This allowed us to assume that it is not only the small NPle size that influenced the increase in the density of NP compacts. In the $CaF_2$ NP used in the present work, there was a certain proportion of metal Ca NPles, as we have previously shown in [24]. Metal NPles could result in reduced interparticle friction during pressing, which could result in the increase in the density of compacts containing NP additives. Note that clusters of calcium NPles were stored in the initial $CaF_2$ NP after annealing the NP in air up to a temperature of 400 °C [24], which is comparable to the temperature of 425 °C achieved during the release of the compact in the MP method. In general, SP compaction tended to increase compaction density with increasing compaction pressure for all samples (except samples 5945 and 5955, Table 3).

Figure 3 shows digital images of 30 compacts produced in this work using two pressing methods—SP and MP.

The pictures in Figure 3 clearly show the change in the color of the compacts depending on the proportion of the NP additive administered. With an increase in the amount of additive, the color of the compacts varies uniformly from white to dark gray. Compacts from pure NP immediately acquired a dark color, from dark gray to black, with an increase in pressing pressure (5957–5959). The maximum blackening was shown by a compact made of pure NP (5970 Table 3) using the MP method with heating to a temperature of 425 °C.

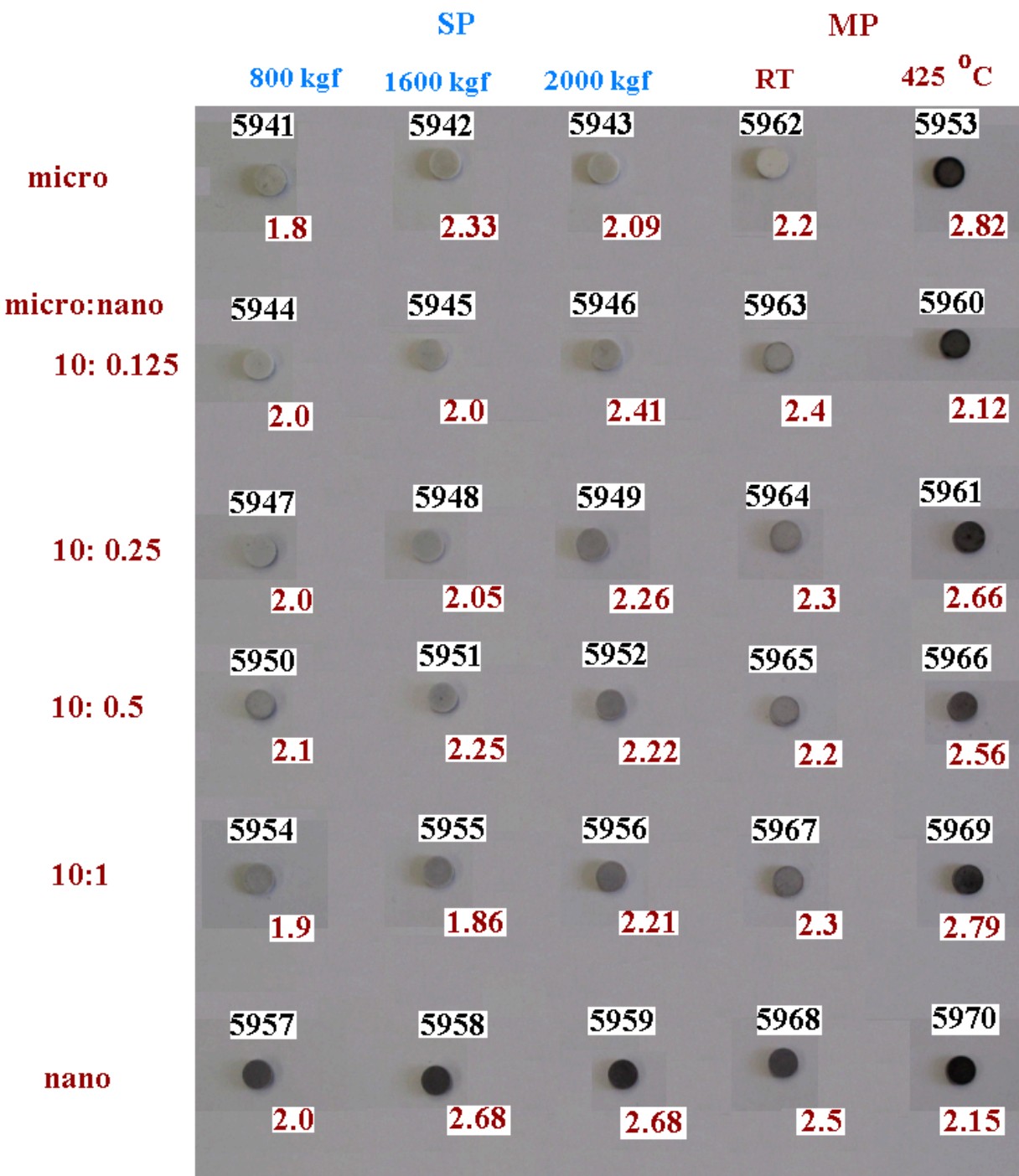

**Figure 3.** Photos of compacts prepared by the MP and SP methods from the mixes of commercial and synthesized CaF$_2$ powders, indicating the density achieved using SP and MP methods. Designation: red numbers—density of compacts (g/cm$^3$).

The black color of all compacts made of pure CaF$_2$ NP (5957–5959, 5968 and 5970 in Figure 3) clearly proves that the blackening of compacts was caused by a high concentration of defects and pores in the NP, which led to an increase in the light absorption band in the visible range and an increase in the reflected black color. Our preliminary experiments on irradiation of CaF$_2$ compacts with high-energy electrons (700 keV) [36] also clearly showed the appearance of black on the surface of the compact from the irradiated side due to the formation of a large number of defects during irradiation.

The density of compacts made from mixtures of micro- and nanopowders of the same chemical composition using different pressing methods, in this case magnetic pulse (MP) and static pressing (SP), depends on a number of factors: texture properties (specific surface area (SSA), size, diameter and total pore volume of powders), structure and phase composition of initial components of mixtures, concentration of NP additive to be added, pressing pressure $T_p$ and compact material temperature during pressing. Let us consider the effects of these factors individually and collectively:

A nonmonotonic change in the compacts' density with an increase in the concentration of NP in the mixtures was observed after SP, regardless of the pressing pressure value. It is likely that such a nonmonotonic change in the density of compacts was determined by the balance of pore and metal Ca NPle concentrations, which had a multi-directional effect on the density of the compacts. A high pore concentration reduces density, and metal Ca NPles act as a "lubricant", which leads to an increase in compacts' density.

The density of the compact made of micron powder by MP method at 425 °C (5953, Table 3) was 2.82 g/cm$^3$ and significantly exceeded the density of the micron powder compact (5942, Table 3) made using the SP method (2.33 g/cm$^3$), which confirms the greater effectiveness of MP compaction compared to the uniaxial one-sided SP method and is consistent with comparative estimates of the effectiveness of various compression methods given in the literature [35,37,38]. Note that the density of sample 5968 (MP, RT) from pure NP was greater (2.57 g/cm$^3$) than that of 5962 (MP, RT) from micron powder (2.26 g/cm$^3$). This proves that the dominant factor in compacting compacts at RT was the concentration of Ca NPles in the compact from NP, and not the concentration of pores.

Maximum density values of compacts made from powder mixtures were also achieved using the MP method, regardless of the concentration of the nanopowder in the compacts, with the exception of sample 5960 ($\rho$ = 2.12 g/cm$^3$), the density of which was lower than that of sample 5946 ($\rho$ = 2.41 g/cm$^3$).

An increase in the temperature during the pressing process to 425 °C led to a further increase in the compaction density of micron powder, from 2.26 to 2.82 g/cm$^3$ (5962 and 5963, respectively), which was quite expected. However, the increase in the pressing temperature had the opposite effect on the density of the pure NP compact (5970), the density of which decreased sharply to 2.15 g/cm$^3$, compared to the density of the pure NP compact (5968, $\rho$ = 2.57 g/cm$^3$) pressed at RT. This decrease in density can also be explained by the oxidation of the metal Ca clusters with an increase in temperature to 425 °C in the MP process, which led to a loss of the "lubrication" effect from the metal Ca particles due to their oxidation to CaO.

The relationship of the change in density of compacts produced by MP (a) and SP (b) to the proportion of NP additive added to the starting powder mixtures is given in Figure 4.

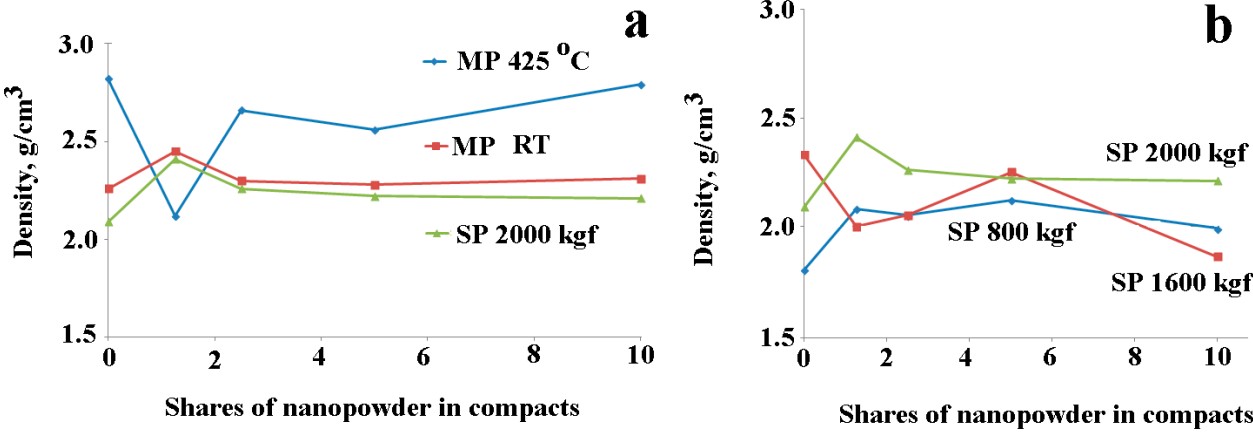

**Figure 4.** Dependence of the density of compacts produced by MP (**a**) and SP (**b**) on the proportion of NP in the initial powder mixtures.

Analysis of dependencies in Figure 4 showed the following:

Compact density (5959, Table 3) from the $CaF_2$ micron powder produced by MP (425 °C) had an absolute maximum (2.82 g/cm$^3$) among 30 samples of compacts, the characteristics of which are given in Table 2, which was 88.7% of the theoretical value of $CaF_2$ density (3.18 g/cm$^3$). Sample 5962 produced by MP at RT had a 20% lower density (2.26 g/cm$^3$) than sample 5959. Thus, increasing the temperature using MP increased the density of samples made of micron powder. A diametrically opposite pattern was observed when pressing samples from pure $CaF_2$ NP (samples 5968 and 5970). The increase in temperature to 425 °C led to a sharp decrease in the density of sample 5970 to 2.15 g/cm$^3$ (67% of theoretical density), while sample 5968, pressed by the MP method without heating, had a density of 2.57 g/cm$^3$, which is comparable to the density of samples produced by the SP method (5958 and 5959) that reached 2.68 g/cm$^3$. A sharp decrease in the density of the compact from pure NP (sample 5970) using MP (425 °C) to a value of 2.15 g/cm$^3$ was caused by the oxidation of metal Ca NPles, which were present in the maximum concentration in the initial $CaF_2$ NP produced by the PEBE method in a vacuum. The formation of additional NPles of calcium oxide from Ca NPles of a smaller size led to an increase in interparticle porosity in the sample and a corresponding decrease in compact density due to the simultaneous passage of two processes—pressing and heating the compact to a temperature of 425 °C. In turn, the absence of phase transformation of Ca $\rightarrow$ CaO in the MP process at RT resulted in a 20% increase in sample density of 5968 ($p$ = 2.57 g/m$^3$) relative to sample density of 5970 ($\rho$ = 2.15). In fact, metal Ca NPles in compact made of pure NP (sample 5968) served as a "lubricant" between $CaF_2$ and Ca NPles, and among themselves, until their oxidation began. Therefore, with pulsed compression, a sufficiently dense compact with a density of 2.57 g/cm$^3$ was produced. This hypothesis is supported by the finding of close values of compact density from mixtures of micro- and nanopowders produced at room temperature by various MP and SP (2000 kgf) methods at RT.

The symbatic behavior of compaction density curves using MP (RT) and SP (2000 kgf) methods is clearly seen in Figure 4a. This behavior of the density curves indicates that compacts of comparable density can be produced using the cheaper SP method compared to MP, when the optimum Ca NPle concentration of the source NP is achieved.

In order to assess the effect of annealing of the starting NP on the properties of compacts made using annealed NP by SP (1600 kgf) and MP pressing at RT and 425 °C, three compacts were made, the density and pressing parameters of which are given in Table 3. $CaF_2$ NP was previously annealed in air at a temperature of 400 °C for 30 min for the purpose of annealing various structural defects which were present at the initial $CaF_2$ NP produced by strong nonequilibrium PEBE method in a vacuum. Previously, we have shown in detail [24] that after annealing at 400 °C, the original plum NP turned bright white, indicating the annealing of most of the structural defects present in the non-annealed NP.

A comparison of the data from Tables 3 and 4 showed that preliminary annealing of NP resulted in a significant increase in the density of the compact pressed from pure NP at 425 °C. The density increased from 2.15 (sample 5970) to 2.84 g/cm$^3$ (sample 5665) and became comparable to the density of the $CaF_2$ micron powder compact (sample 5953, $\rho$ = 2.82 g/cm$^3$). However, compacts' densities pressed at RT from annealed pure NP were slightly lower than compacts' densities made at RT from unheated NP by MP (2.39 g/cm$^3$ (sample 5670, Table 4) and 2.57 g/cm$^3$ (5968, Table 3)) and SP at the pressure of 16.57 g/cm$^3$.

**Table 4.** Density of $CaF_2$ compacts made from $CaF_2$ NP annealed at 400 °C and parameters of their pressing by SP and MP methods.

| Sample | Composition | Degassing | | Pressing | | Release | | Compact | | | |
|---|---|---|---|---|---|---|---|---|---|---|---|
| | | $T_d$, °C | $t_d$, min | $s_p$ ($U_c$, kV) | $T_p$, °C | $T_r$, °C | $t_r$, min | m, g | Ø, mm | d, mm | ρ, g/cm³ |
| 5663 | $CaF_2$ | RT | - | 1600 kgf | RT | - | - | 0.065 | 4.63 | 1.52 | 2.54 |
| 5665 | $CaF_2$ | 425 | 120 | 0.8 | 425 | 425 | 60 | 0.071 | 4.55 | 1.54 | 2.84 |
| 5670 | $CaF_2$ | RT | 30 | 0.8 | RT | - | - | 0.046 | 4.64 | 1.14 | 2.39 |

Designations: RT—room temperature; $s_p$—static pressure; $t_d$—degassing time; $t_r$—release time; $U_c$—charging voltage; $T_d$, $T_p$ and $T_r$—degassing, pressing and release temperature, respectively. Compact options: m—mass, Ø—diameter, d—thickness, ρ—density.

*3.3. Pulsed Cathodoluminescence (PCL) Analysis*

Cathodoluminescence of crystals is one of the important elements of spectral analysis of solids and allows the determination of the internal structure of the sample (chemical composition, presence of defects, impurities, internal stresses, etc.).

The PCL method [32–34] is an express method of luminescent analysis of condensed substances at their temperatures. PCL is excited by high-current pulse electron beams (up to 180 keV), and its intensity is much greater than the intensity of other known types of luminescence. By means of PCL, it is possible to determine the mineral appearance of the analyzed sample, the presence of intrinsic defects and impurities in it, the coordination position of the impurity ions and their charging state, and much more without destroying the sample itself. Currently, methods of analysis of various minerals based on PCL are actively being developed [32,37,38].

PCL spectra of compacts of pure micro- and nanopowders and mixtures thereof recorded at room temperature are shown in Figures 5–7. Note that PCL spectra of all compacts were recorded separately in the UV and Vis spectrum regions, and all spectra (except PCL spectra of samples produced by MP at 425 °C) were two wide bands with maxima at ~315 and 500 nm. Typical examples of such spectra are the PCL spectra in the UV-Vis sample area from Table 4 (5663, 5665 and 5670), which are shown in Figure S1 (Supplementary Materials). Note the good agreement of our PCL spectra with the cathodoluminescent (CL) spectra given in [39]. CL spectra in [39] showed two broad maxima at 290 nm and 550 nm. The first band in the UV portion of the spectrum was associated with structural defects such as F center, H center, I center, anion vacuum and/or Vk center ($F^{2-}$ molecular ion), and the second band in the Vis region of the spectrum was associated with point defects associated with the $Mn^{2+}$ impurity in the nodes of the cationic lattice $CaF_2$ [39]. Considering that the influence of the pressing parameters (pressure, temperature) had practically no effect on the location of the maxima or on the form of PCL spectra of compacts in the UV region of the spectrum (PCL spectra in the UV region of all compacts from Table 3 are given in Figure S2 (Supplementary Materials)), we excluded this spectrum region from consideration and focused only on the analysis of the Vis region of the spectrum.

In Figure 5a, PCL spectra in the Vis area of compacts from $CaF_2$ micron powder are given. The intensity of PCL compacts produced by the SP method (5941–5943) decreased uniformly with increasing pressing pressure. The compact density and intensity of PCL of sample 5962, obtained by MP at RT, were comparable to the corresponding characteristics of samples 5941–5943. Spectra of samples 5941–5943 and 5962 contained one wide peak in the wavelength range of ~350–700 nm.

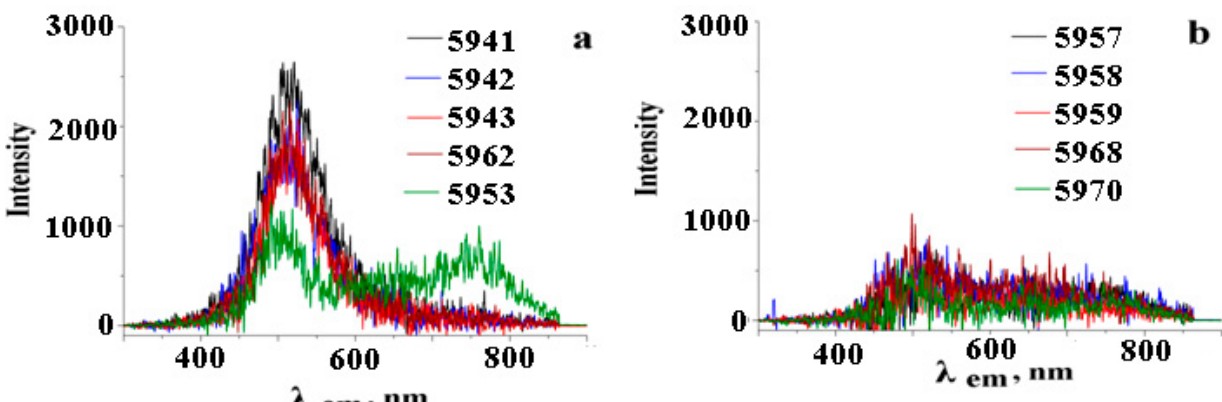

**Figure 5.** PCL spectra of compact specimens 5941–5943, 5962 and 5963 (**a**) and 5957–5959, 5968 and 5970 (**b**) in the visible wavelength range.

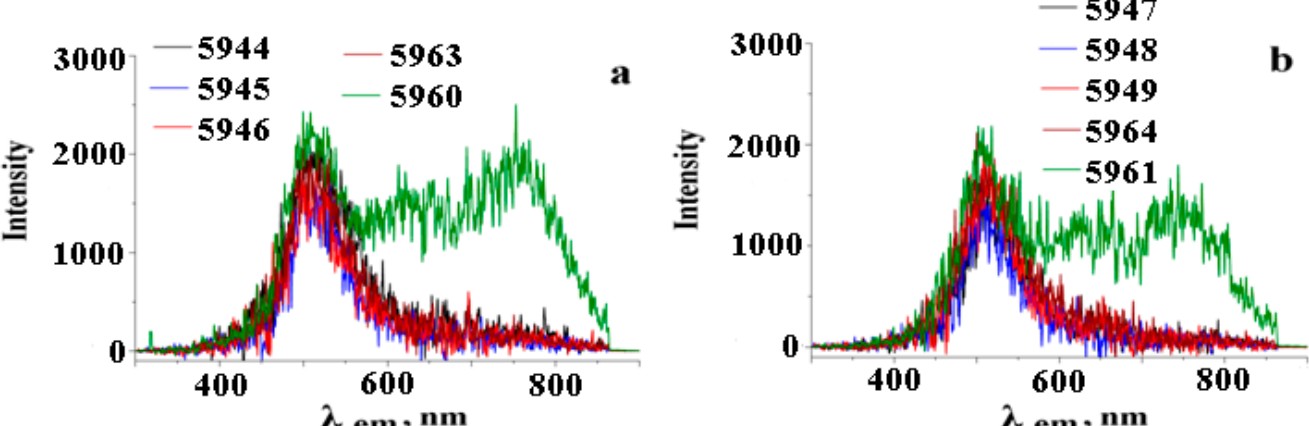

**Figure 6.** PCL spectra of compact specimens 5944–5946, 5963 and 5960 (**a**) and 5947–5949, 5964 and 5961 (**b**) in the visible wavelength range.

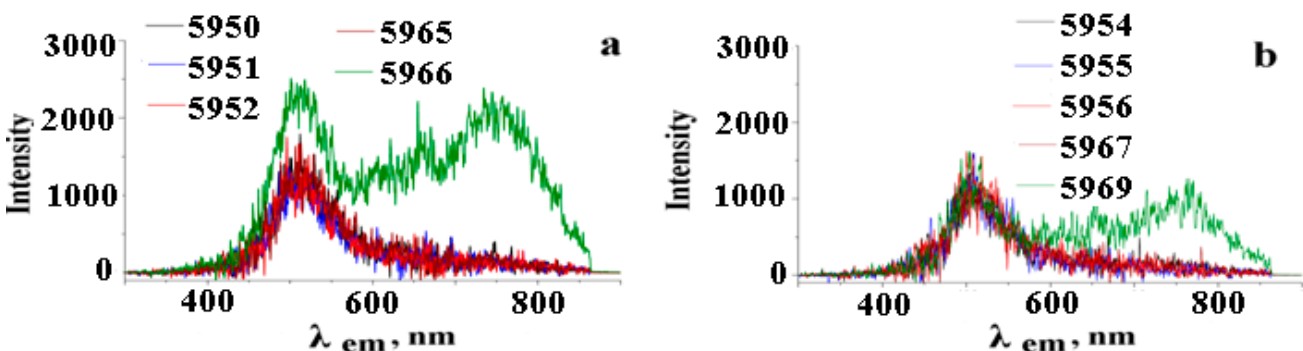

**Figure 7.** PCL spectra of compact specimens 5950–5952, 5965 and 5966 (**a**) and 5954–5956, 5967 and 5969 (**b**) in the visible wavelength range.

However, sample 5353 (MP at pressing temperature $T_p$ = 425 °C) had a more complex spectrum form, consisting of three wide peaks with highs at wavelengths of ~500, 650 and 700 nm. Looking ahead, the PCL spectra of all NP compacts produced by MP at Tp = 425 °C contained three broad peaks with maxima at ~500, 650 and 700 nm. Accordingly, the PCL spectra of all the samples produced by MP at RT contained only one broad peak with a maximum at 500 nm, similar to the spectra of the PCL samples produced by SP.

In Figure 5b, the ranges of PCL of the compacts of clean $CaF_2$ NP made by the MP and SP methods are given. It can be seen that the samples of the compacts were practically not

luminescent in the Vis wavelength range. However, in the absence of pressing, pure $CaF_2$ NP showed PCL (see Figure 7a, curve 2, and Figure 7c in work [24]). The low luminescence intensity of compacts 5957–5959, 5968 and 5970 can be explained by the high concentration of reduced metallic calcium NPles in the composition of NP produced under vacuum [24], which led to the concentration suppression of luminescence.

Figures 6 and 7 show PCL spectra in the visible wavelength range of samples made from mechanical mixtures of micro- and nanopowders in proportions of 10:0.125, 10:0.25, 10:0.50 and 10:1.

The evolution of spectra with an increase in the concentration of nanopowder in the samples showed the following:

(a) The increase in the NP content of the mixtures resulted in a uniform reduction in PCL intensity of samples produced by SP and MP methods at RT (Figure 6), which was quite expected.

(b) The density of compacts (or pressing pressure) produced using SP and MP methods at room temperature did not significantly affect the intensity of PCL spectra (Figures 5–7) relating to a particular composition.

(c) The intensity of PCL compacts pressed at RT was significantly affected by the HP—the intensity decreased with the increase in the proportion of NP additive in the mechanical mixtures (Figures 6 and 7).

(d) The main factor affecting the morphology of the PCL spectra of all luminescent samples (including slightly luminescent in the visible wavelength range samples 5957–5959, 5968 and 5970 of pure NP $CaF_2$) was the pressing temperature $T_p$ using the MP method with heating of the samples to 425 °C.

It is well known that $O_2$ diffusion into the crystal cell is observed when $CaF_2$ is heated in air. The anion $O^{2-}$ diffuses deep into the crystal from the surface. The possibility of partial substitution of fluorine for oxygen is associated with the proximity of ion radii, electronic configurations and electronegativity of fluorine and oxygen atoms [40]. From the PCL spectra curves in Figure 5a, Figure 6a,b and Figure 7a,b, it can be seen that heating the compacts to a temperature of 425 °C resulted in a long-wave arm in the PCL spectra of $CaF_2$ micron powder (5963) and samples from powder mixtures (5960, 5961, 5966 and 5961), which contained two wide overlapping bands in the red (~630 nm) and near-infrared (NIR) (~750 nm) ranges. The appearance of two additional peaks in samples containing NP was possibly due to the formation of CaO oxide, as a result of oxidation of Ca NPles upon heating to a temperature of 425 °C.

The appearance of a glow in the red/NIR region of the spectrum can also be associated with carbon contamination of the samples at a temperature of 425 °C, which is partly confirmed by the dark color of all samples produced by the MP method at 425 °C. However, the dark color of all samples (5957–5959, 5968 and 5970) produced by MP (at RT and 425 °C) and SP (800–2000 kgf, RT) from pure NP did not result in a significant intensity of the PCL peak at 750 nm. In Figure 3, in the picture of sample 5963, traces of graphite contamination on the end surface of the compact are clearly visible (graphite was used as a lubricant in compacting compacts and was applied to the inner surface of the mold), but this did not lead to the appearance of a red strip in the spectrum of this sample. Therefore, factors such as Ca NPles or carbon contamination can be excluded from consideration when explaining the origin of PCL peaks at 630 and 750 nm.

Thus, it is apparent that the pressing temperature is a major factor that has produced a red band in the PCL spectrum of compacts produced by MP at 425 °C. Temperature and high pressure during pressing of the compacts could lead to a change in the phase composition of compacts from powder mixtures, oxidation of metal Ca NPles and growth of nano- and microcrystals as a result of recrystallization, but this is unlikely for a compact from micron powder. Therefore, the most likely, in our opinion, consequence of the effect of temperature on the compacts is the formation of vacancies in the $CaF_2$ fluoride lattice as a result of the removal of fluorine atoms from the surface layers of the lattice. It is the appearance of fluorine vacancies that leads to the appearance of annealed red band

samples and NIR bands with highs at 630 and 750 nm in the spectra. This conclusion is consistent with the conclusions of [41], which shows the influence of substrate temperature on $CeF_3$ thin films prepared by the thermal evolution method. Analysis of the composition of the films confirmed the formation of cerium oxyfluoride, which led to the formation of free fluorine ions. Our early studies [24] also showed that in the PCL spectra of non-annealed $CaF_2$ NP (sample S0) and $CaF_2$ NP annealed in air at relatively low temperatures of 200 and 400 °C (samples S200 and S400), three peaks were present: two green peaks (498 nm and 524–544 nm) and a red peak (627–678 nm).

There was no peak in the NIR region at 760 nm in the above NP samples; however, after annealing NP at a high temperature of 900 °C, a powerful wide NIR peak appeared in the sample spectrum with a maximum at a wavelength of 713 nm, which further confirms that the nature of the NIR band in the compacts is associated with fluoride vacancies and the formation of the oxyfluoride phase.

The appearance of NIR peaks in samples produced by MP at RT and 425 °C is solely due to the addition of $CaF_2$ NP; the matrix of $CaF_2$ micron powder itself is not associated with the appearance of peaks in the NIR spectrum region. An NIR peak with an intense maximum at 713 nm was detected after annealing $CaF_2$ NP in air at a temperature of 900 °C in our earlier work [24].

In compacts (samples 5663, 5665 and 5670) produced by SP and MP methods from pure $CaF_2$ NP pre-annealed in air at a temperature of 400 °C, the dominant peak was in the red region of the spectrum (599–611 nm), but there was no third peak in the NIR region. This fact indicates that pre-annealing of $CaF_2$ NP at relatively low temperatures (300–400 °C) allows eliminating most of the defects contained in the initial $CaF_2$ NP produced under vacuum conditions by the PEBE method. Obviously, pre-annealing of defects in the original NP can produce compacts with densities greater than those of compacts of non-annealed NPs. This statement is supported by compacts 5656, 5665 and 5670 in Table 4. Interestingly, the intensity ratio of red (dominant) to green (minor) band of $I_{red}/I_{green}$ in compacts 5656, 5665 and 5670 increased by 3–6 times compared to $I_{red}/I_{green}$ in compacts 5959, 5968 and 5970 produced from non-annealed powders under the same pressing conditions.

Thus, PCL spectra analysis showed that the addition of $CaF_2$ NP to the matrix of $CaF_2$ micron powder allows a controlled change in the location of the dominant peak of the matrix, shifting it into either the shortwave or the longwave region using different pressing methods. Using the MP method with a compact heating of up to 425 °C, an intense (up to 50% of the intensity of the dominant peak) NIR peak appears in PCL spectra, which can be controlled by simple thermal annealing of the compact in air.

With a compact density of 89% achieved, it is quite possible to expect to obtain transparent ceramics after annealing such a compact in a vacuum, without using high-temperature pressing methods, which will be tested by us in the future.

## 4. PL Analysis

In order to more accurately detect the nature of defects in compacts of mixtures of $CaF_2$ micro- and nanopowders, we used a well-known structurally sensitive photoluminescence (PL) method. PL spectra of the 30 samples previously shown in Figure 3 were recorded at room temperature and are shown in Figure 8.

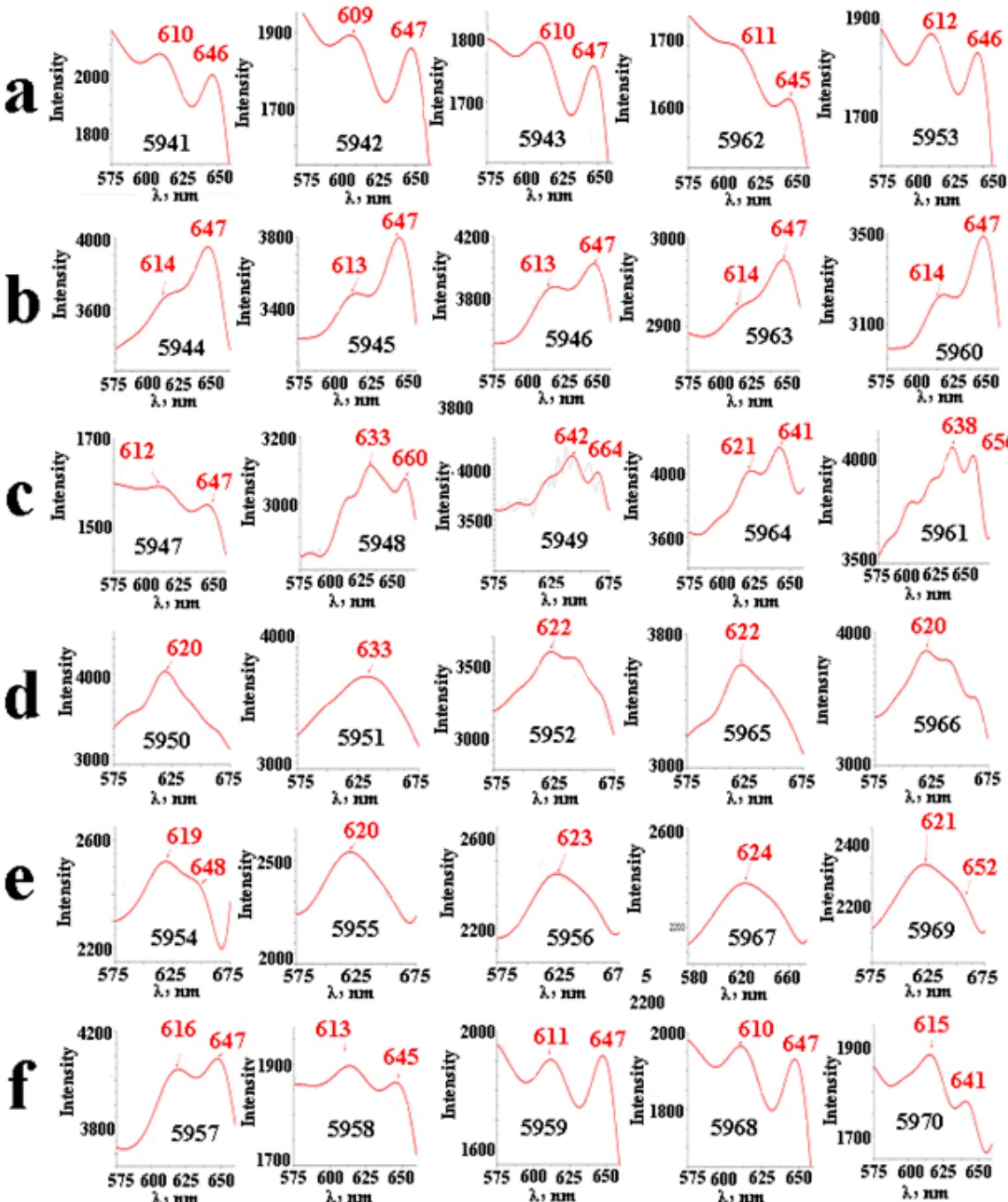

**Figure 8.** PL spectra of compact specimens in the visible wavelength range: 5941–5943, 5962, 5953 (**a**); 5944–5946, 5963, 5960 (**b**); 5947–5949, 5964, 5961 (**c**); 5950–5952, 5965, 5966 (**d**); 5954–5956, 5967, 5969 (**e**); 5957–5959, 5968, 5970 (**f**). Thick red line—smoothed PCL spectrum.

Compacts of micron powder (5941–5943, 5962, 5953) had the same morphology. The spectra had two red bands with maxima at 609–611 and 645–647 nm, regardless of the pressing method. The PL intensity decreased uniformly (5941–5943) with increasing pressing pressure in the SP method at RT. Sample 5962, pressed at higher pressure by MP at RT, maintained the same downward trend in PL intensity. In turn, sample 5953 (MP at 425 °C) showed an increase in the intensity of both PL bands compared to sample 5962, which could be caused by an increase in crystal size at 425 °C.

Compacts from clean NP (samples 5957–5959) with similar spectra to the corresponding spectra of samples 5941–5943 showed a sharper decrease in PL intensity with an increase in pressure in the SP method.

Compacts 5968 and 5970 compacted by the MP method showed an opposite tendency of decreasing intensity with increasing pressing temperature, which can be explained by the influence of the metal Ca particles contained in the initial $CaF_2$ NP produced under vacuum conditions.

The type of spectra in the samples with a small fraction of the nanoadditive (1.25) was maintained, as was that in the compacts of micron powder, regardless of the pressing method. The spectra contained two red bands with maxima at 613–614 and 645–647 nm. With an increase in the fraction of the nanoadditive in compacts, the spectra changed greatly.

At the same time, a high intensity of PL with peaks at 621–638 and 656–664 was shown by all samples with a nanodopant concentration of 2.5 compared to sample 5947, in which the third band was clearly absent.

With an increase in the share of nanodopant up to 5–10, regardless of the pressing method, the structure of spectra changed towards the smoothing of the peaks. There was a gradual transformation of the shape of the spectra into a wide band with one maximum at about 619–624 nm, although the shape of the bands clearly indicated the presence of a third peak at 656–664 nm. Peaks with maxima at ~640 and 660 nm could be observed on the spectra of individual samples (5952, 5966, 5954 and 5969).

The peaks of compacts with nanodopant fractions of 5–10 using the SP method were uniformly reduced with an increase in pressing pressure, which indirectly indicated a decrease in the number of defects in the compacts. With MP compacts, a significant decrease in PL intensity, regardless of the pressing temperature, occurred with an increase in the nanodopant fraction from 5 to 10. Spectra of compacts produced by MP at 425 °C (5966 and 5969) had a more pronounced form than those of the RT compacts, which were simpler.

In general, the most intense were samples 46 and 49, pressed at a maximum pressure of 2000 kg/s with minimum fractions of nanodopant (1.25–2.5). In turn, the maximum intensity was also shown by samples 5964 and 5961 manufactured by the MP method with a nanodopant fraction of 2.5, which allows us to consider the above fraction as optimal for obtaining compacts with a maximum yield of PL.

The appearance of a blue peak (434 nm) was observed in the PL spectra of compacts (5663 and 5665) pressed from $CaF_2$ NP pre-annealed at 400 °C (Figure 9a).

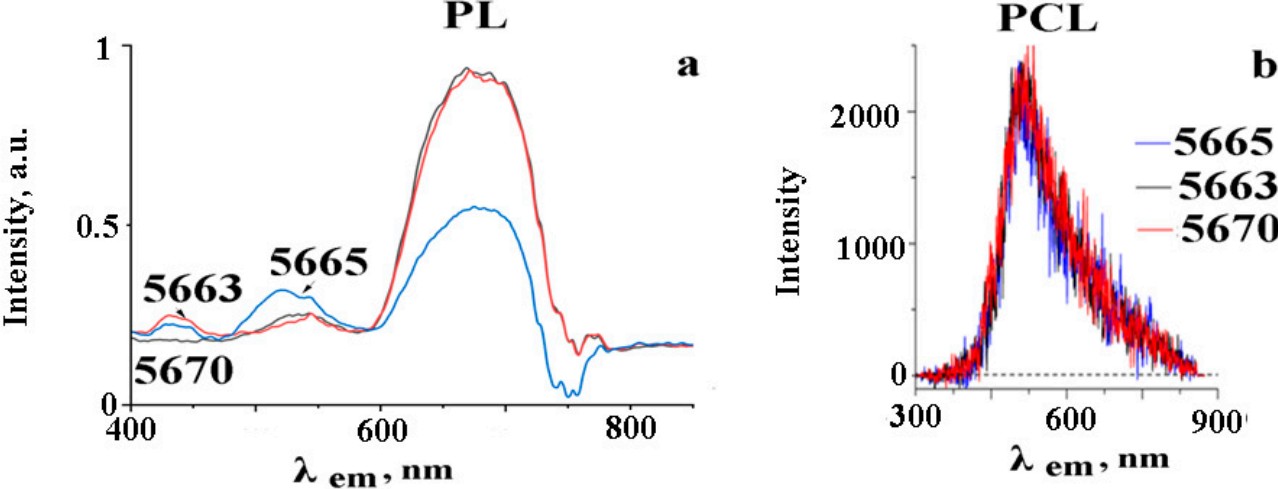

**Figure 9.** (**a**) PL spectra of compacts 5663, 5663 and 5670; (**b**) PCL spectra of compacts 5663, 5663 and 5670.

The blue band was absent from the PCL spectra of samples 5663 and 5665 (Figure 9b). It is most likely that the appearance of the blue peak is associated with an impurity oxygen vacancy in the nanocrystalline $CaF_2$ lattice [42–44].

## 5. Conclusions

$CaF_2$ compacts from mixtures of micro- and nanopowders were produced by SP and MP methods. We have demonstrated that the features of the original nanopowder dopant, such as the concentration and defective structure of the dopant, as well as the pressing temperature have a significant effect on the density of the compacts. The density of pure NP compacts produced by SP and MP at room temperature reached 84% of the theoretical value, and the density of the pure micron powder compact produced by MP at 425 °C was 89%. Preliminary annealing of the initial NP at a temperature of 400° also made it possible to obtain a compact with a density of 89% of the theoretical value. The density of compacts from mixtures of nano- and micropowders, regardless of the method of pressing at room temperature, did not exceed 76% of the theoretical density; however, pressing by MP at a temperature of 425 °C made it possible to achieve a compact density of 88%. The strong correlation between the properties of the additive, $CaF_2$ NP (oxygen vacancies, the concentration of Ca NPles in the initial NP), in compacts and cathode-photoluminescent spectra of compacts was shown. The addition of the NP dopant to the micron powder $CaF_2$ matrix allows controlled displacement of the dominant peak in the PCL spectra of compacts, into either the shortwave or the longwave region, using different pressing techniques. With MP, upon heating the compact up to 425 °C, an intense (up to 50% of the intensity of the dominant peak) NIR peak always appeared in PCL spectra, which can be controlled by subsequent thermal annealing of the compact in air. In the PL spectra of compacts, in addition to two to three intense bands in the red spectrum (610–660 nm), an additional peak in the blue band was found, probably associated with an impurity oxygen vacancy.

The preparation of dense $CaF_2$ compacts (up to 89% of theoretical density) indicates the need for further studies of their sintering ability in a vacuum in order to assess the possibility of obtaining clear ceramics from nanopowders of alkaline-earth and rare-earth fluorides synthesized by the PEBE method earlier [24–26].

**Supplementary Materials:** The following supporting information can be downloaded at: https://www.mdpi.com/article/10.3390/photonics9100782/s1, Figure S1: PCL spectra of samples of compacts from CaF2 (5663, 5665 and 5670) in the visible region of the spectrum (a) and in the UV range (b). Figure S2: PCL spectra of samples of compacts from CaF2 in the UV range (SP- static pressing, MP- magnetic pulse pressing).

**Author Contributions:** Conceptualization, V.G.I. and S.Y.S.; methodology, V.G.I.; investigation, S.V.Z. and M.G.Z.; writing-original draft preparetion, V.G.I. and M.G.Z.; project administration, S.Y.S. All authors have read and agreed to the published version of the manuscript.

**Funding:** The reported study was funded by Russian Science Foundation, project number 22-19-00239.

**Institutional Review Board Statement:** Not applicable.

**Informed Consent Statement:** Not applicable.

**Data Availability Statement:** Not applicable.

**Acknowledgments:** The authors are grateful to the junior researcher of the Laboratory of Pulsed Processes of the Institute of Electrophysics Ural Branch of the Russian Academy of Sciences, K.I. Demidova, for textural analysis. The authors are grateful to the staff of the IEP Ural Branch of the Russian Academy of Sciences; A.V. Spirina for recording PCL; junior researcher O.R. Timoshenkova for SEM images of NPs; and M. Rähne, Institute of Physics, University of Tartu (Estonia), for HRTEM images of NPs.

**Conflicts of Interest:** The authors declare no conflict of interest.

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
