# Peer review of "Properties of Compacts from Mixtures of Calcium Fluoride Micro- and Nanopowders"

_photonics, doi:10.3390/photonics9100782_

Round 1

Reviewer 1 Report

In the paper under review, the authors investigated the compaction of micro- and nano-powders  or calcium fluorides by cold static pressing or by pressing under magnetic pulse action. The commercial CaF2 reagent was used as starting micropowders. Nanopowders were prepared by original technique. The mixture of micro- and nano-powders were used as precursors also. The protocols of synthesis and characterization, and results of measurements, are described in details.

It was found that the density of prepared pellets was very low in all cases. The color of the pellets varies from grey to black. Some special luminescence bands, connected with point defects, have been revealed.

The results of this work correspond well with the conclusion from the papers [7,8]: the quality of CaF2 ceramics is determined by the quality of precursor. The precursors were used in this work are unsuitable for the preparation of optical ceramics.

Publication of this paper is possible after minor revision.

1.    The reference 2 and 3 should be replaced with the papers, connected with oxygen laser ceramics.

2.    The English of the paper should be improved.

3.    The typos should be corrected.

For example, page 2, Inroduction, line 9: “In hot forming …” must be replaced with “In hot pressing…”

Author Response

Dear editors of the Journal of Photonics

The authors are grateful to the reviewers for their interest to our work and their time spent for reading and studying it. We tried to take into account all the comments of reviewer and gave detailed answers to all his comments and questions. See red text, please.

Respectfully, from the team

Ph.D. Vladislav Ilves

14-10-2022

Comments and Suggestions for Authors

In the paper under review, the authors investigated the compaction of micro- and nano-powders  or calcium fluorides by cold static pressing or by pressing under magnetic pulse action. The commercial CaF2 reagent was used as starting micropowders. Nanopowders were prepared by original technique. The mixture of micro- and nano-powders were used as precursors also. The protocols of synthesis and characterization, and results of measurements, are described in details.

It was found that the density of prepared pellets was very low in all cases. The color of the pellets varies from grey to black. Some special luminescence bands, connected with point defects, have been revealed.

The results of this work correspond well with the conclusion from the papers [7,8]: the quality of CaF2 ceramics is determined by the quality of precursor. The precursors were used in this work are unsuitable for the preparation of optical ceramics.

Publication of this paper is possible after minor revision.

  1. The reference 2 and 3 should be replaced with the papers, connected with oxygen laser ceramics.

Answer: The reference 2 and 3 have been replaced

  1. The English of the paper should be improved.

Answer: The English was tested by a native speaker

  1. The typos should be corrected.

For example, page 2, Inroduction, line 9: “In hot forming …” must be replaced with “In hot pressing…”

Answer:  Replaced

Reviewer 2 Report

Ilves  et al. present the properties of CaF2 compacts from mixtures of micro and nano powders were produced by SP and MP methods. There are, however, some comments for the authors.

Major comments:

1.       On page 4, the authors said the smoothing was applied. The detailed smoothing process must be provided.

2.       In Figure 7, the authors claim there are two peaks, 630 nm and 750 nm. How do you define those peaks? Especially, the one at 630nm, the statement looks arbitrary.

3.       Figure 8 shows the PL spectra of the samples. It is really hard to see the raw PL spectra. Please make them more distinguishable. Also, there are peaks that authors didn’t mention. For example, 5949 shows peaks at ~ 585 nm and 625 nm.

4.       How did authors measure PCL and PL? Since authors were using absolute intensities of spectra to evaluate the quality of the samples. The sample preparation is important.

Minor comments:

1.       Page 4, “… lamp. Photoelectronic multiplier R928 from..” It should be photomultiplier tube.  

2.       Page 4, “electron beam with a duration of 2 ns at…” Duration should be the pulse temporal width. If the duration here doesn’t mean pulse temporal width, please clarify in the context.

3.       Table 1, what do the numbers in parentheses mean?

4.       Table 2, the standard deviation of the average pore diameter is required.

5.       Table 3 can be improved. Authors can rename the sample like SP5957. It is really hard for readers to get the sample information from Table 3.

6.       Page 9, “Fig. 3. Photos of compacts prepared…” to Fig. 3. Photo of compacts prepared…

7.       Page 13, Figure 5. PCL spectra of compact specimens: 5941-5943, 5962, 5963 (a), There is no 5963 on the figure.

8.       Page 14, “Fig. 6.7 shows PCL spectra in the visible wavelength…” to Fig. 6 and 7 shows PCL spectra in the visible wavelength

Author Response

Dear editors of the Journal of Photonics

The authors are grateful to the reviewers for their interest to our work and their time spent for reading and studying it. We tried to take into account all the comments of reviewer and gave detailed answers to all his comments and questions. See red text, please.

Respectfully, from the team

Ph.D. Vladislav Ilves

14-10-2022

Comments and Suggestions for Authors

Ilves  et al. present the properties of CaF2 compacts from mixtures of micro and nano powders were produced by SP and MP methods. There are, however, some comments for the authors.

Major comments:

  1. On page 4, the authors said the smoothing was applied. The detailed smoothing process must be provided.

To eliminate noise in the spectra, they were smoothed in the Origin program in a known way using FFT filter (Fourier filtering) for 100-150 points.

  1. In Figure 7, the authors claim there are two peaks, 630 nm and 750 nm. How do you define those peaks? Especially, the one at 630nm, the statement looks arbitrary.

We did a three-peak deconvolution (Origin program) of all samples 5953, 5960, 5961, 5966,5969 and 5970). Convergence was achieved for all PCL spectra of the above samples. The results of deconvolution were not given in order not to clutter up the article with redundant information. In particular, the location of the average peak in the article is indicated approximately (~ 630), since it varied in the range (610 -638 nm). The position of the peak at  ~750 nm changed similarly.

  1. Figure 8 shows the PL spectra of the samples. It is really hard to see the raw PL spectra. Please make them more distinguishable. Also, there are peaks that authors didn’t mention. For example, 5949 shows peaks at ~ 585 nm and 625 nm.

On fig. 8 there are no original spectra. We removed them at the request of one of the reviewers. The simultaneous presence of the original and smoothed spectrum greatly "overloads" Figure 8.

The unmentioned peaks ( ~ 585 nm and 625 nm) are of relatively low intensity and therefore their wave length was not specified by the authors.

  1. How did authors measure PCL and PL? Since authors were using absolute intensities of spectra to evaluate the quality of the samples. The sample preparation is important.

The PL spectra were recorded on MDR-204 spectrometers (deuterium lamp, KLM-H980-200-5 laser module, Hamamatsu PMT R928). Radiation from a deuterium lamp was directed through a filter onto a sample fixed in a holder. The PL spectrum of the sample was recorded with a spectrometer.

The samples were tablet-shaped compacts (Figure  3). Therefore, there were no problems with recording photo and cathodoluminescence spectra.

Minor comments:

  1. Page 4, “… lamp. Photoelectronic multiplier R928 from..” It should be photomultiplier tube.

 Fixed, thanks

  1. Page 4, “electron beam with a duration of 2 ns at…” Duration should be the pulse temporal width. If the duration here doesn’t mean pulse temporal width, please clarify in the context.

The pulse characteristics of the electron nanosecond accelerator RADAN were provided to us directly by the developer of the KLAVI-1 spectrometer. The pulse duration is 2 nanoseconds.

  1. Table 1, what do the numbers in parentheses mean?

The measurement error in the last decimal place is given in parentheses.

Fixed, thanks

  1. Table 2, the standard deviation of the average pore diameter is required.

Unfortunately, the standard deviation of the mean pore diameter in a Micromeritics TriStar 3000 V6.03 A setup is not estimated.

  1. Table 3 can be improved. Authors can rename the sample like SP5957. It is really hard for readers to get the sample information from Table 3.

Fixed, thanks

Subscripts sp and mp for sample numbers are given only in Table 3 for clarity.

  1. Page 9, “Fig. 3. Photos of compacts prepared…” to Fig. 3. Photo of compacts prepared…

Fixed, thanks

  1. Page 13, Figure 5. PCL spectra of compact specimens: 5941-5943, 5962, 5963 (a), There is no 5963 on the figure.

We have corrected the error in the caption to the picture. Correct-sample 5953

Fixed, thanks

  1. Page 14, “Fig. 6.7 shows PCL spectra in the visible wavelength…” to Fig. 6 and 7 shows PCL spectra in the visible wavelength

Fixed, thanks